# Variants rs3804099 and rs3804100 in the *TLR2* Gene Induce Different Profiles of TLR-2 Expression and Cytokines in Response to Spike of SARS-CoV-2

**DOI:** 10.3390/ijms252011063

**Published:** 2024-10-15

**Authors:** Julio Flores-González, Zurisadai Monroy-Rodríguez, Ramcés Falfán-Valencia, Ivette Buendía-Roldán, Ingrid Fricke-Galindo, Rafael Hernández-Zenteno, Ricardo Herrera-Sicairos, Leslie Chávez-Galán, Gloria Pérez-Rubio

**Affiliations:** 1Laboratory of Integrative Immunology, Instituto Nacional de Enfermedades Respiratorias Ismael Cosio Villegas, Mexico City 14080, Mexico; juliofglez@gmail.com (J.F.-G.); ricardo13herrera@gmail.com (R.H.-S.); 2HLA Laboratory, Instituto Nacional de Enfermedades Respiratorias Ismael Cosío Villegas, Mexico City 14080, Mexico; zurimonroe2@gmail.com (Z.M.-R.); dcb_rfalfanv@hotmail.com (R.F.-V.); ifricke@iner.gob.mx (I.F.-G.); 3Translational Research Laboratory on Aging and Pulmonary Fibrosis, Instituto Nacional de Enfermedades Respiratorias Ismael Cosío Villegas, Mexico City 14080, Mexico; ivettebu@yahoo.com.mx; 4COPD Clinic, Instituto Nacional de Enfermedades Respiratorias Ismael Cosío Villegas, Mexico City 14080, Mexico; rafherzen@yahoo.com.mx

**Keywords:** SARS-CoV-2, TLR-2, genetic variants, COVID-19, rs380499, rs3804100

## Abstract

The present study aimed to identify in patients with severe COVID-19 and acute respiratory distress syndrome (ARDS) the association between rs3804099 and rs3804100 (*TLR2*) and evaluate the expression of TLR-2 on the cell surface of innate and adaptive cells of patients’ carriers of C allele in at least one genetic variant. We genotyped 1018 patients with COVID-19 and ARDS. According to genotype, a subgroup of 12 patients was selected to stimulate peripheral blood mononuclear cells (PBMCs) with spike and LPS + spike. We evaluated soluble molecules in cell culture supernatants. The C allele in *TLR2* (rs3804099, rs3804100) is not associated with a risk of severe COVID-19; however, the presence of the C allele (rs3804099 or rs3804100) affects the TLR-2 ability to respond to a spike of SARS-CoV-2 correctly. The reference group (genotype TT) downregulated the frequency of non-switched TLR-2+ B cells in response to spike stimulus; however, the allele’s C carriers group is unable to induce this regulation, but they produce high levels of IL-10, IL-6, and TNF-α by an independent pathway of TLR-2. Findings showed that TT genotypes (rs3804099 and rs3804100) affect the non-switched TLR-2+ B cell distribution. Genotype TT (rs3804099 and rs3804100) affects the TLR-2’s ability to respond to a spike of SARS-CoV-2. However, the C allele had increased IL-10, IL-6, and TNF-α by stimulation with spike and LPS.

## 1. Introduction

Severe acute respiratory syndrome coronavirus 2 (SARS-CoV-2) is the causative agent of coronavirus disease 2019 (COVID-19) [1]. The spectrum of SARS-CoV-2 infection can vary from asymptomatic to severe. Regarding severe disease, patients need hospitalization, intensive care unit (ICU) support, and mechanical ventilation, and complications can arise, including acute respiratory distress, multi-organ dysfunction, and even death [2].

In the lungs of patients with severe COVID-19, as opposed to mild cases, aberrant recruitment of inflammatory macrophages and infiltration of T lymphocytes, including cytotoxic T cells and neutrophils, have been measured [3].

Toll-like receptor 2 (TLR-2) is mainly involved in sensing different pathogens, including bacteria, viruses, fungi, and parasites. This is required for an inflammatory response during β-coronavirus infection; it can sense the envelope (E) protein of SARS-CoV-2 to initiate inflammatory signaling pathways and cytokine production [4]. In addition, the spike (S) protein of SARS-CoV-2 is a potent viral pathogen-associated molecular pattern (PAMP) sensed by TLR-2 in lung epithelial cells [5]. Subsequently, TLR-2 forms heterodimers with TLR-1 or TLR-6, creating a complex containing MyD88 with IRAK kinase family members, leading to activation of NF-κB and MAPK signaling, and ultimately to the production of inflammatory cytokines and chemokines [6]. The expression of TLR-2 increases following SARS-CoV-2 infection, and it is positively associated with the severity of COVID-19 [4].

The TLR-2 encoding gene (*TLR2*) is located on the long arm of chromosome 4 (4q32), where several single-nucleotide polymorphisms (SNPs) have been identified. Multiple studies have assessed the role of rs380499 and rs3804100 in infectious diseases and cancer. Patients with chronic gastritis carriers of CT or TT genotypes (rs3804099) show a more severe degree of systemic inflammation by the increased level of circulating neutrophils and C-reactive protein [7]. Variant rs3804099 results in synonymous polymorphism (Asn199Asn); in in silico analysis, it shows that C > T nucleotide variation has a 70% probability of affecting *TLR2* mRNA splicing, while rs3804100 (C > T) results in Ser450Ser; however, it shows activation of an additional splice site [8].

In in vitro assays, *TLR2* mRNA expression in peripheral blood mononuclear cells (PBMCs) with stimulation by peptidoglycan (PGN) showed that carriers with the TT genotype in rs3804099 had higher expression compared with individuals carrying the CC or CT genotype [9]. PBMCs from healthy subjects with the CT/TT genotypes at *TLR2* rs3804099 produced higher amounts of TNF-α, IL-1β, and IL-6 after the *Legionella pneumophila* stimulation [10]. Patients with the cryptococcal meningitis carrier of CT genotype (rs3804099) had low levels of proinflammatory cytokines (L-1β, IL-1α, IL-6, IL-8, IL-15, TNF-α, and IFN-γ) and chemokines (eotaxin, monocyte chemoattractant protein [MCP]-1, macrophage inflammatory protein [MIP]-1α, and MIP-1β) in cerebrospinal fluid [11]. In leprosy patients, carriers of the T allele (rs3804099) produced higher levels of IL-17 and IL-6 [12].

The present study aimed to identify the association between the genetic variants rs3804099 and rs3804100 in the *TLR2* gene in COVID-19 severe patients who have acute respiratory distress syndrome (ARDS) and to evaluate the expression of TLR-2 on the cell surface of innate and adaptive cells of patients’ carriers of the C allele in selected SNPs.

## 2. Results

### 2.1. Study Population

Both groups have no statistically significant difference in age, sex, and body mass index (BMI); however, patients with severe COVID-19 and ARDS showed more days of hospital stay (21 vs. 18, *p* < 0.001); 100% of patients with severe ARDS required invasive mechanical ventilation (IMV) and needed more days of using this life support (16 vs. 12 *p* < 0.001). There was a higher percentage of patients who died in those with severe ARDS (49.6 vs. 30.7, *p* < 0.001) (Table 1).

### 2.2. Genotype, Allele, and Haplotype Association in Variants of TLR2

Both genetics variants comply with the Hardy–Weinberg equilibrium (rs3804099; *p* = 0.068 and rs3804100; *p* = 0.299). There was no significant association between the study groups at the genotype and allele level (Appendix A). The haplotype analysis shows that rs3804099–rs3804100 (Appendix A) had high linkage disequilibrium (r^2^ = 93%); however, there is no associated haplotype in the study population (Appendix A). The data availability is presented in the ClinVar repository, submission number SUB14513466.

### 2.3. COVID-19 Patients’ Carriers of the Allele C Do Not Modify the Frequency of TLR-2+ Monocytes with Spike Stimulus

Monocytes were identified by the CD14 and HLA-DR expression and absence of CD2 and CD3 (Figure 1A). We found that the frequency of TLR-2+ monocytes is not modified in patients with the C allele in *TLR2* (rs3804099 or rs3804100), and it is not altered even when stimulated with a spike or spike + LPS stimulus (Figure 1B). The TLR-2 expression intensity does not show changes (Appendix A).

### 2.4. COVID-19 Patients of the Reference Group Decreased the TLR-2 Frequency in Non-Switched B-Cells

Based on the expression of CD27 and IgD, circulating total B cells were categorized into naïve, non-switched, and switched subsets (Figure 2A), and the TLR-2 expression was examined. The total B cells TLR-2+ frequency did not show differences (Appendix A). Similarly, the frequency of TLR-2+ in naïve and switched B cells was not modified (Figure 2B and Figure 2D, respectively). However, the frequencies of the TLR-2+ non-switched subset decreased in COVID patients with TT genotypes (reference group) when they were stimulated with a spike (*p* = 0.0076) and spike + LPS (*p* = 0.0499), both compared to the unstimulated condition. The TLR-2 expression intensity into naïve, non-switched, and switched B-lymphocytes was not modified (Appendix A). Blue squares for the reference group and red circles for the allele C carriers’ group.

Next, TLR-2 expression in activated B cell subsets (CD69+ B cell) was evaluated (Figure 3A). COVID-19 patients with the TT genotypes (rs3804099 and rs3804100) decreased the frequency of naïve TLR-2+CD69+ B cells under spike + LPS stimulus (*p* = 0.0148) (Figure 3B), non-switched TLR-2+CD69+ B cells by spike stimulus (*p* = 0.0207) (Figure 3C), and switched TLR-2+CD69+ B cells with spike + LPS stimulus (*p* = 0.0286) (Figure 3D) compared to the unstimulated condition. The TLR-2 expression intensity (Appendix A) showed that COVID-19 patients with the C allele showed high TLR-2 expression compared with patients’ carriers of TT genotypes (rs3804099 and rs3804100) in naïve active B cells (*p* = 0.0485) (Appendix A). Non-switched or switched B active cells do not show changes (Appendix A).

### 2.5. The C Allele to TLR2 Does Not Affect the Frequency of TLR-2+ Cytotoxic T Cells

The expression of TLR-2 was evaluated by flow cytometry in CD3+CD8+ T-cells (Figure 4A) and activated CD8+ T-cells (CD69+, Figure 4C), as well as the TLR-2 expression intensity (Appendix A). It was compared between COVID-19 patient groups. Our data showed that the TLR-2+CD8+ T-cell frequency under the stimulus conditions was not altered (Figure 4B), similar to the TLR-2 expression intensity (Appendix A). However, the TLR-2+CD69+CD8+ T cell subset frequency from patients with TT genotypes decreased under spike + LPS stimulus (Figure 4D), and the intensity of expression was not modified (Appendix A). Other cytotoxic cells, such as NK and NKT, were also evaluated by flow cytometry, but the frequency of these subpopulations did not show differences.

### 2.6. PBMCs from COVID-19 Patients with the C Allele Have a Low Capacity to Produce Inflammatory Cytokines by Spike Stimulus

Cytokines and soluble proteins were evaluated in the cell culture supernatant. PBMCs from COVID-19 patients with the C allele stimulated with spike protein do not show cytokine production. However, cells from COVID-19 patients with the C allele significantly increase cytokine production when stimulated with spike + LPS, mainly IL-10, IL-6, and TNF-α compared to the unstimulated condition (Table 2 and Figure 5).

Both COVID-19 patients produced FasL efficiently when stimulated with a spike. Other soluble cytotoxic molecules did not show changes. Patients with the T allele do not show a significant change in cytokine production. (Appendix A).

## 3. Discussion

We included patients with COVID-19 and ARDS with demographic and clinical data similar to other study cohorts. The principal risk factors for the severity of COVID-19 have been described, including age (60 years old), sex (more than 65% of the patients included were men), and comorbidities [13]. Patients with severe ARDS need more days in the hospital, showed low PaO_2_/FiO_2_, and a higher proportion of death in comparison to COVID-19 with mild to moderate ARDS. All patients with severe ARDS need IMV.

TLRs are important in recognizing viral particles and activating the innate immune system. The TLR-1/TLR-2 heterodimer binding SARS-CoV-2 S protein contributes to the inflammatory state and lung injury seen in COVID-19 [14]. TLR-2 is the innate sensor that triggers β-coronavirus-induced inflammatory cytokine expression [4]. Studies suggest *TLR2* gene variations can alter immune responses to pathogens and disease outcomes.

We found no significant association between ARDS risk in COVID-19 and rs3804099 and rs3804100 in *TLR2*. Our findings, which align with a previous report on the Brazilian population, suggest that the *TLR2* rs3804100 variant may not be a significant factor in developing symptomatic COVID-19 [15]. However, a study on the Egyptian population reported that the C allele of rs3804099 was more frequent in patients with COVID-19 than in people without the disease (OR = 2.42) [16]. These results could have implications for future research on population-specific susceptibility to COVID-19.

When we analyzed the TLR-2 expression, we observed that the presence of the C allele in rs3804099 or rs3804100 does not affect the expression of TLR-2 evaluated by flow cytometry (FMI) in monocytes, B cell subpopulations, or cytotoxic T cells. We do not see FMI changes even in stimulated conditions with spike or LPS in patients with COVID-19. Similar results were observed in the frequency of monocytes and CD8+ T cells, even in activation states about allele C carriers’ group.

Regarding B cells, the allele C carriers’ group did not change the frequency of the non-switched B-cells TLR-2+ subset compared to the reference group and was independent of the stimuli. However, it is important to note that this subset changes in the reference group; this means that when cells are stimulated with the spike protein, the frequency of non-switched TLR-2+ B cells is decreased, data that agree with a previous report where authors demonstrated that after the interaction spike/TLR-2, the receptor is internalized [17].

We speculate that patients who carry the C allele have affected this internalization process, which is crucial for developing long-term immunity. Evidence demonstrated that after the interaction of the TLR-2/viral ligand, the TLR2 internalization triggers IRF7/3 or IRF2/IRF1/STAT1 signaling to upregulate IFN-β and IFN-α, respectively [18]. Probably, those patients with the C allele in rs3804099 or rs3804100 SNPs have affected this immune mechanism. Thoughtful studies are necessary to clarify the internalization of TLR-2 in the context of COVID-19 infection. However, patients of the stimulated reference group had lowered TLR-2 expression on non-switched B cells or activated B cells and in cytotoxic T cells. Previously, it was reported that stable populations of memory B cells are likely generated following SARS-CoV-2 infection and that these memory B cells correlate to an effective primary response [17].

This study did not evaluate the production of cytokines by cell population; however, quantification of proinflammatory cytokines in supernatant showed that peripheral blood leukocytes in response to LPS and spike stimulation of patients with C allele (rs3804099 or rs3804100) were significantly associated with increased production of IL-10, IL-6, and TNF-α. These data corroborate the results previously observed where the cytokine production was significantly increased in patients with the C allele (rs3804099) compared with those with the T allele in response to LPS stimulation [17]. This finding suggests that the C allele affects the TLR-2 ability to respond to a spike correctly.

This study has some limitations. We only included patients with severe COVID-19 and did not evaluate the production of cytokines by cell population. In the in vitro assays, we should have used a stimulus more in line with TLR-2 that would allow us to differentiate whether it was forming heterodimers with TLR-1 or TLR-6. TLR-2 can recognize microorganisms through different domains, and SARS-Cov-2 variants have differential affinity between receptors (mainly TLR-4), which influences signaling to a greater or lesser extent [19]. Reports indicate that SNPs of *TLR2* affect protein expression; however, whether they affect differential binding to SARS-CoV-2 variants is still unknown, which is undoubtedly an excellent field for future studies. According to our results, the genetic variants rs3804099 and rs3804100 affect the mechanism of immunological response by SARS-CoV-2.

## 4. Materials and Methods

### 4.1. Study Population

We included 1018 hospitalized patients with COVID-19 at the Instituto Nacional de Enfermedades Respiratorias Ismael Cosío Villegas (INER) in Mexico. Characteristics of the patients were described previously [20]. The individuals were invited to participate and signed informed consent (approbation codes: C53-20, B11-23), and we complied with the Helsinki Declaration. The patients were classified according to ARDS into two categories: severe (PaO_2_/FiO_2_ ≤ 100 mmHg) and mild-moderate (PaO_2_/FiO_2_ > 100 mmHg) [21].

### 4.2. DNA Extraction, Quantification, and Genotyping

Peripheral blood was obtained in tubes with ethylenediaminetetraacetic acid (EDTA). The extraction of genetic material was described previously [20]. Genotyping of rs3804099 and rs3804100 was performed by real-time polymerase chain reaction using the StepOnePlus device (Applied Biosystems, Foster City, CA, USA) by allelic discrimination through predesigned TaqMan probes (C__22274563_10 (rs3804099) and C__25607727_10 (rs3804100) Applied Biosystems). Amplification conditions were described previously [20].

### 4.3. In Vitro Assays

A subgroup of 12 patients was selected according to the genotype in rs3804099 and rs3804100 (Appendix A). We classified into two subgroups: patients’ carriers with TT genotypes (n = 8) in both SNPs (reference group) and patients carrying at least one allele C in one or two SNPs (n = 4, allele C carriers group). PBMCs were isolated from 15 mL of the blood by standard Lymphoprep ^TM^ (Accurate Chemical-Scientific, Westbury, NY, USA). The PBMCs were then cryopreserved until use, ensuring their integrity for subsequent experiments by using the trypan blue dye as a cell stain to determine PBMC viability. PBMCs were seeded in 24-well, flat-bottomed cell culture plates (Corning Costar Sigma-Aldrich, St Louis, MO, USA) at a density of 5 × 10^5^ cells/well in an RPMI 1640 medium supplemented with two mM L-glutamine, 1M HEPES (Gibco™, Paisley, Scotland, UK), an antibiotic–antimycotic solution of Penicillin–Streptomycin–Amphotericin B (Gibco™ Paisley, Scotland, UK), and 10% fetal bovine serum (Gibco™ Paisley, Scotland, UK). The cell cultures were maintained for 24 h at 37 °C in a controlled, humidified atmosphere containing 5% CO_2_, ensuring optimal conditions for their growth and function. The stimulation with SARS-CoV-2 Spike Protein S1/S2 (BioLegend, San Diego, CA, USA), LPS + Spike, and untreated (only culture media) was described previously [20]. LPS induces signaling through a TLR-2-independent pathway, and this stimulus is frequently used as a model for in vitro polyclonal activation [22]. In our model, the LPS stimulus is a control to distinguish between antigen-specific signaling (TLR-2/spike) and signaling independent of TLR-2 and the antigen.

### 4.4. Flow Cytometry Staining and Soluble Molecules Evaluation

After culture, cells were collected for flow cytometry staining. The cells were incubated with an antibody cocktail for 30 min at RT to identify monocytes (CD14, CD2, CD3, HLA-DR), B lymphocytes (CD19, IgD, IgM, CD27, CD69, HLA-DR), and cytotoxic cells including CD8 T-cell, NK and NKT cells (CD3, CD8, CD56, CD69), and CD4 T-lymphocytes (CD3, CD4, HLA-DR, CD38). All populations were marked with anti-TLR-2 antibodies. To establish a baseline for comparison, the cells used for the Fluorescence Minus One (FMO) condition were stained and acquired in parallel. This allowed us to identify background levels of staining; dead cells were omitted using viability staining (Appendix A). Finally, the cells were washed, suspended in cell staining buffer (BioLegend), and kept at 4 °C until acquisition. The data were acquired using a FACS Aria II flow cytometer (BD Biosciences, San Jose, CA, USA) equipped with the FACSDiva 6.1.3 software (BD Biosciences, San Jose, CA, USA). In each condition, at least 50,000 events of the interested population were acquired per sample. The flow cytometry data file (FCS) was analyzed using Flow Jo (Flow Jo, LLC, Ashland, OR, USA)™ v10.6.1. Supernatants from culture assays were stored at −70 °C until use. Cytokines and cytotoxic molecules were measured using a human CD8/NK panel (13 plex) from BioLegend, following the provided manufacturer’s instructions (Appendix A).

### 4.5. Statistical Analysis

We applied Mann–Whitney U tests to quantitative variables. Categorical variables were analyzed with contingency tables and χ2 with Yates’ correction. We used SPSS v.21.0 (IBM Corp, Armonk, NY, USA. Released, 2011) to describe the demographic and clinical variables of the study population or Epidat v.3.1 software (Xunta de Galicia and Panamerican Health Organization, 2006). The association of genotypes and alleles was made using Epidat v.3.1 software. We calculated Hardy–Weinberg equilibrium (HWE) using p^2^ + 2pq + q^2^ = 1, where p is the frequency of the “A” allele, and q is the frequency of the “a” allele in the population [23]. The haplotype analysis was performed in Haploview 4.2 software [24]. After Skewness and Kurtosis in the data, the Shapiro–Wilk test was selected as an appropriate method of normality test. According to the normal distribution results, plots show median and IQR or mean and SD. Each plot used a Kruskal–Wallis’s test with Dunnett’s post-test for multiple comparisons test for the comparison in stimulation conditions. In contrast, we used the Mann–Whitney U test to compare the groups between conditions. (GraphPad Software V 10.2.3, Inc., San Diego, CA, USA). For cytometry analysis, comparisons between conditions were made; however, no significant differences were found with the sample size studied.

## 5. Conclusions

Genetic variants in *TLR2* (rs3804099, rs3804100) are not associated with a risk of severe COVID-19; however, the presence of the C allele (rs3804099 or rs3804100) affects the TLR-2’s ability to respond to SARS-CoV-2 spike protein correctly. In addition, our findings showed that TT genotypes (rs3804099 and rs3804100) affect the non-switched TLR-2+ B cell distribution.

## Figures and Tables

**Figure 1 ijms-25-11063-f001:**
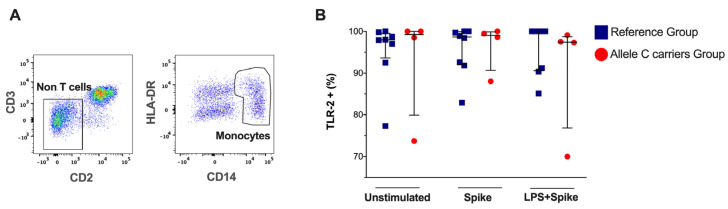
TLR-2 expression in monocytes of patients with COVID-19 according to genotypes (rs3804099 and rs3804100). They were stimulated for 24 h with spike protein (1 µg/mL) or spike + LPS (1 µg/mL). An unstimulated condition was included as a control stimulation (Unstimulated). (**A**) Representative dot plots show the limitation of CD2-D3−, then the gate CD14+HLA-DR+. (**B**) The frequency of TLR -2+ monocytes is reported, and each dot represents an independent patient. Data are expressed as median and IQR values. The statistical comparisons were performed using the Kruskal-Wallis test. Blue squares are used for the reference group, and red circles for the allele C carriers’ group.

**Figure 2 ijms-25-11063-f002:**
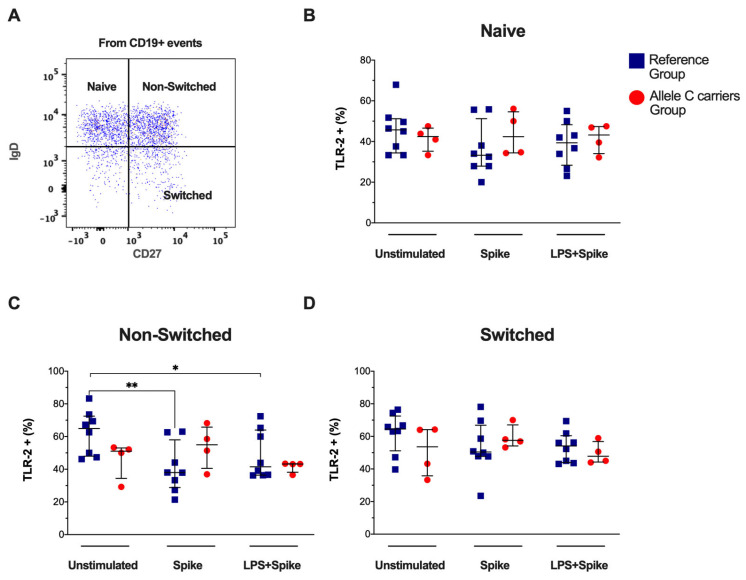
Spike decreased TLR-2 frequency in non-switched B-cells subset by patients with the TT genotypes (rs3804099 and rs3804100). Mononuclear cells from two groups of patients were stimulated for 24 h with spike protein (1 µg/mL) or spike + LPS (1 µg/mL). An unstimulated condition was included as a control stimulation (Unstimulated). (**A**) Representative dot plots show the B -cells subsets distribution based on CD27 and IgD expression as naive (IgD + CD27 −), non-switched (IgD + CD27+), or switched (IgD -CD27+). Analysis of TLR-4+ B -cells subsets frequencies for (**B**) naïve, (**C**) non-switched, and (**D**) switched. Data were represented as median and IQR values. The Kruskal -Wallis test performed statistical comparisons, * *p* < 0.05, ** *p* < 0.01. Blue squares are used for the reference group, and red circles for the allele C carriers’ group.

**Figure 3 ijms-25-11063-f003:**
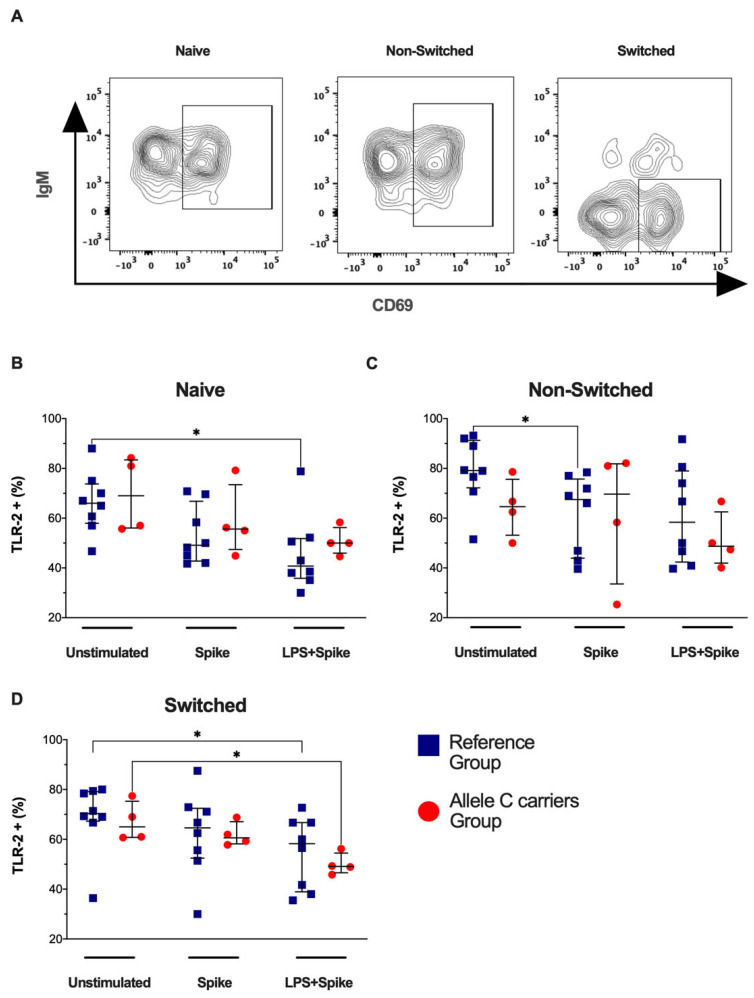
Activated B-cell subsets do not change the frequency of TLR-2+ when stimulated. Mononuclear cells from two groups of patients were stimulated for 24 h with spike protein (1 µg/mL) or spike + LPS (1 µg/mL). An unstimulated condition was included as a control stimulation (unstimulated). (**A**) Representative dot plots show the distribution of activated B-cell subsets based on IgM and CD69 expression. The frequency of activated B-cell subsets positive to TLR-4 was analyzed; thus, TLR-4+ in (**B**) naïve, (**C**) non-switched, and (**D**) switched are shown. Data are represented as median and IQR values. The Kruskal–Wallis test performed statistical comparisons, * *p* < 0.05. Blue squares are used for the reference group, and red circles for the allele C carriers’ group.

**Figure 4 ijms-25-11063-f004:**
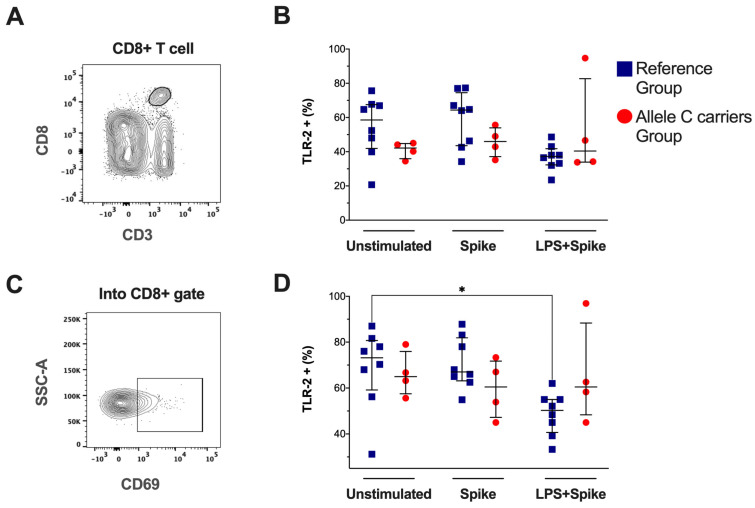
The TLR-2 frequencies in activated CD8-T cells decreased in patients with the TT genotypes (rs3804099 and rs3804100). Mononuclear cells from two groups of patients were stimulated for 24 h with spike protein (1 µg/mL) or spike + LPS (1 µg/mL). An unstimulated condition was included as a control stimulation (unstimulated). (**A**) Representative dot plots show the CD8+ T-cell based on CD3 and CD8 expression. (**B**) Frequency of total CD8+ T-cell and TLR-4+CD8+ T cell. (**C**) Representative dot plots show the CD69 expression in the CD8+ T-cell gate. (**D**) Frequency of CD69+CD8+ T-cell, TLR-4+CD69+CD8+ T cell. Data were represented as median and IQR values. The Kruskal-Wallis test performed statistical comparisons, * *p* < 0.05. Blue squares are used for the reference group, and red circles for the allele C carriers’ group.

**Figure 5 ijms-25-11063-f005:**
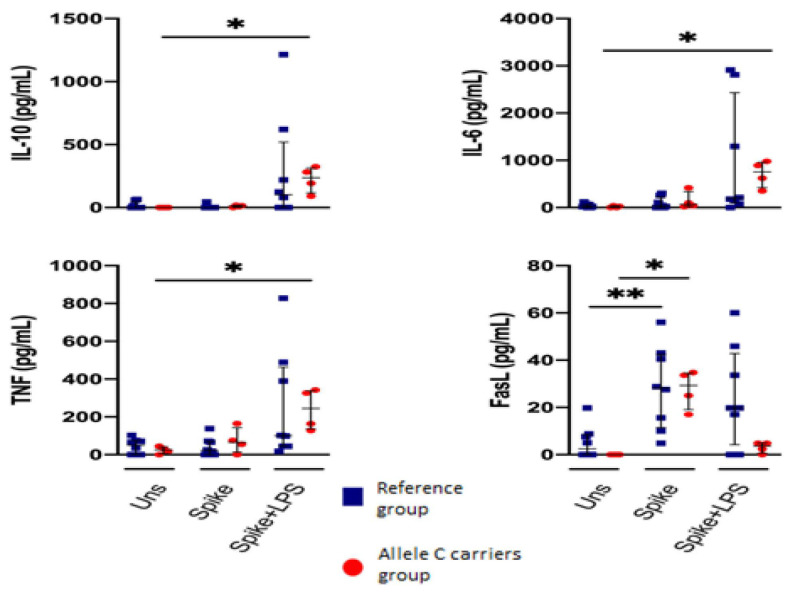
The C allele (rs3804099 or rs3804100) does not modify the secretion of inflammatory or cytotoxic cytokines by PBMCs. Mononuclear cells from two groups of patients were stimulated for 24 h with spike protein (1 µg/mL) or spike + LPS (1 µg/mL each one). An unstimulated condition was included as a control stimulation (unstimulated). The cytotoxic LEGENDplex^TM^ panel assessed culture supernatants for nine protein markers. Data are represented as median and IQR values. The Kruskal–Wallis test performed statistical comparisons, * *p* < 0.05, ** *p* < 0.01. Blue squares are used for the reference group, and red circles for the allele C carriers’ group.

**Table 1 ijms-25-11063-t001:** Demographic and clinical data of patients with COVID-19 and ARDS.

Variable	ARDS Severe (*n* = 413)	ARDS Mild-Moderate (*n* = 605)	*p*-Value
Age (years)	59 (51–68)	58 (49.5–67)	0.270
Male, n (%)	294 (71.2)	413 (68.3)	0.355
BMI (kg/m^2^)	30 (27–34)	29 (26–33)	0.166
Hospital stay (days)	21 (15–35)	18 (11.5–28)	6.0 × 10^−7^
PaO_2_/FiO_2_ (mmHg)	73 (61–86)	174 (136–216)	2.2 × 10^−16^
IMV, n (%)	413 (100.0)	367 (60.6)	<0.001 *
Days with IMV	16 (9–27)	12 (0–22)	3.7 × 10^−10^
Deceased, n (%)	205 (49.6)	186 (30.7)	<0.001 *

ARDS: Acute Respiratory Distress Syndrome. BMI: Body Mass Index. PaO_2_: partial pressure of arterial oxygen. FiO_2_: fraction of inspired oxygen. IMV: Invasive Mechanical Ventilation. Mann–Whitney U test was used to *p*-value. * Contingency tables were used to calculate the *p*-value for variables male, IMV, and deceased.

**Table 2 ijms-25-11063-t002:** Cytokines evaluated in PBMCs from COVID-19 patients with C allele (rs3804099 or rs3804100).

Cytokine pg/mL	Spike + LPS	Unstimulated	*p*-Value
IL-10	237 (115–313)	0 (0–0)	0.0286
IL-6	756 (420–957)	12 (0–34)	0.0286
TNF-α	245 (137–338)	25 (4–41)	0.0286

We show median values and interquartile ranges (25–75). The Kruskal–Wallis test performed statistical comparisons.

## Data Availability

Data availability statements are available in ClinVar number SUB14513466, https://www.ncbi.nlm.nih.gov/clinvar/?term=SUB14513466 (accessed on 17 March 2024).

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
