# Peer review of "Variants rs3804099 and rs3804100 in the TLR2 Gene Induce Different Profiles of TLR-2 Expression and Cytokines in Response to Spike of SARS-CoV-2"

_ijms, 2024, doi:10.3390/ijms252011063_

Round 1
Reviewer 1 Report
Comments and Suggestions for Authors
It is possible that having certain unique SNPs may alter resistance to COVID and make the patient more susceptible to severe disease. If this were known, it would be possible to predict in advance how severe a patient's disease would become by looking for it during testing. The authors investigated the two using patient cells. However, not so many differences were observed. So, the title is rather overstated and the abstract ‘Findings showed 27 that TT genotypes (rs3804099 and rs3804100) affect the non-switched TLR-2+ B cell distribution.’ nor was the rationale clearly stated. The tone would need to be changed.
Table 1 P-values are evidences in themselves, so please give specific values, not <0.001. We imagine that it was probably much smaller.
2.2 Table S1, 2
It would be easier to understand if the figures for healthy people were known as controls here, how many people have C? It would seem that being hospitalised, whether severely ill/not severely ill, is special enough. This is an area that we would like to see investigated.
Table 2 What are the data on patients with T allele? Shouldn't T and C be compared here?
In this sense, S6 should be shown here.
The authors are probably applying a test of statistics on a brute force basis, which is likely to lead to multiplicity of tests. It should be avoided. 
The main focus of this paper should be on the difference between T and C. So what has to be tested is the difference between T and C under the same conditions. But for example, there is no such indication for LPS+spike or unstimulated in Fig. 2C. isn't it meaningless to compare unstimulated and +spike? And are these differences between C and T not an important requirement for this paper to be valid or not (although I believe from the figures that there would be significant differences between them)? Please make those clear.
Minor, comply with the Hardy-Weinberg equilibrium (rs3804099; 90 p=0.068 and rs3804100; p=0.299). On what basis is this derived? Strictly speaking, isn't Hardy-Weinberg equilibrium only possible if at least a few generations are observed? I would say that the observed proportions of T and C and TT, TC and CC are consistent.
Also see Figure 1.B
The distribution clearly deviates from the normal distribution, which makes the median and IQR unreliable, probably because the cells used in Figs 2 and 3 are diverse, causing large differences. In any case, it is possible that the IQR does not represent the state of the data at present.
Calculate the percentage of each data drop from 100 per cent and take log log10(100-TLR-2)
This would probably normalise the data quite well, and a QQ-normal plot would confirm this.
I am not sure whether I am allowed to ask the authors for the following; I leave it to the authors to decide.
I suspect that the reason for the less significant differences observed here may lie in the way the data were compiled: it may have been a mistake to use qualitative data on the expression of TLR-2 as to whether it is present or not. In general, converting quantitative data into qualitative ones reduces sensitivity.
From Fig. 1 to Fig. 4, we are looking at the results of 12 samples, where the on/off ratio is plotted for each sample, resulting in n=8 or n=4. In fact, as far as Fig. S2A shows, the authors used flow cytometry. If we used this data and did a violin plot with measurements instead of on/off, the differences would be very clear. If you review the data in this way, you may come to a different conclusion. I would try this view at least once.
Author Response
We appreciate the time spent reviewing this paper and the comments made for its improvement.
It is possible that having certain unique SNPs may alter resistance to COVID and make the patient more susceptible to severe disease. If this were known, it would be possible to predict in advance how severe a patient's disease would become by looking for it during testing.
- The authors investigated the two using patient cells. However, not so many differences were observed. So, the title is rather overstated and the abstract ‘Findings showed 27 that TT genotypes (rs3804099 and rs3804100) affect the non-switched TLR-2+ B cell distribution.’ nor was the rationale clearly stated. The tone would need to be changed.
Dear reviewer, As you rightly point out, our study did not reveal significant differences. In light of this, and after careful consideration of the study's limitations, we have chosen to adjust the title (Variants rs3804099 and rs3804100 in the TLR2 gene induce different profiles of TLR-2 expression and cytokines in response to spike of SARS-CoV-2) and abstract (lines 28-31).
2. Table 1 P-values are evidences in themselves, so please give specific values, not <0.001. We imagine that it was probably much smaller.
We appreciate the comment.
We have changed the p values to be exact.
However, in the p values obtained using a contingency table, the program Epidat only allows up to 5 decimals.
We also use epiinfo (https://www.cdc.gov/epiinfo/esp/es_pc.html), which provides up to 8 decimals.
Using both programs, the value of p is very small, so the value of <0.001 remains.
Next, we used GraphPad, and the p-value is very small.
3. 2 Table S1, 2
We show p-value with three decimals
4. It would be easier to understand if the figures for healthy people were known as controls here, how many people have C? It would seem that being hospitalized, whether severely ill/not severely ill, is special enough. This is an area that we would like to see investigated.
We included 1,018 patients hospitalized with COVID-19; we did not include healthy people in this study because it was a third-level hospital. Line 262
5. Table 2 What are the data on patients with T allele?. Shouldn't T and C be compared here?
Lines 188-189. Thank you for the observation. We add the sentence: “Patients with the T allele do not show a significant change in cytokine production”. See Supplementary Figure S6.
6. In this sense, S6 should be shown here.
Thank you for the observation. We have changed the figure S6 to the main article (figure 5)
7. The authors are probably applying a test of statistics on a brute force basis, which is likely to lead to multiplicity of tests. It should be avoided.
Thank you for your comments and concerns. We selected Kruskal–Wallis’s and Dunn's multiple comparisons tests for the comparison in stimulation conditions. In contrast, we used the Mann–Whitney U test to compare the groups between conditions.
This is clarified in the method section (lines 325-327).
8. The main focus of this paper should be on the difference between T and C. So what has to be tested is the difference between T and C under the same conditions. But for example, there is no such indication for LPS+spike or unstimulated in Fig. 2C. isn't it meaningless to compare unstimulated and +spike? And are these differences between C and T not an important requirement for this paper to be valid or not (although I believe from the figures that there would be significant differences between them)? Please make those clear.
The comparison exercise between conditions was performed using the Mann-Whitney U test; however, although there are apparent differences between groups, these are not significant, probably because the sample size is insufficient. This information has been inserted in the text Lines 324-327.
9. Minor, comply with the Hardy-Weinberg equilibrium (rs3804099; 90 p=0.068 and rs3804100; p=0.299). On what basis is this derived? Strictly speaking, isn't Hardy-Weinberg equilibrium only possible if at least a few generations are observed? I would say that the observed proportions of T and C and TT, TC, and CC are consistent.
Line: 320-321. We used the equation: p2 + 2pq + q2 = 1; where p is the frequency of the "A" allele, and q is the frequency of the "a" allele in the population. The equation is an expression of the principle known as Hardy-Weinberg equilibrium, which states that the amount of genetic variation in a population will remain constant from one generation to the next in the absence of disturbing factors. [PMID: 28374190, 32231685].
10. Also see Figure 1.B. The distribution clearly deviates from the normal distribution, which makes the median and IQR unreliable, probably because the cells used in Figs 2 and 3 are diverse, causing large differences. In any case, it is possible that the IQR does not represent the state of the data at present.
Following your suggestion, the updated version now includes the assessment of the normality of data using Skewness as a measure of symmetry and Kurtosis as a measure of the peakedness of a distribution. Furthermore, the Shapiro–Wilk test was selected as an appropriate method for small sample sizes. The distribution in all plots is shown as median and IQR or mean and SD. We added this information in lines 322-324.
11. Calculate the percentage of each data drop from 100 per cent and take log log10(100-TLR-2). This would probably normalise the data quite well, and a QQ-normal plot would confirm this.
Your comment is assertive; however, using the proposed normalization, the reported unit (normalized data) is low (range 1–1.8), making the biological phenomenon challenging to interpret. From our viewpoint, a better alternative is to normalize using the absolute number. However, this would require extensive time because we would need to request authorization from the committee to access medical records and obtain the total leukocyte count. Therefore, we kindly ask for your understanding in allowing us to maintain the report in percentages.
12. I am not sure whether I am allowed to ask the authors for the following; I leave it to the authors to decide.
Thank you for this relevant point. However, the results show frequency, the proportion of cells expressing a marker. This will vary between donors. Regarding expression, we use florescent intensity, a relative measure of how much protein a cell expresses. We selected the median because it is a midpoint of the population (middle channel). This is the preferred method to measure the MFI of a logarithmic histogram.
13. I suspect that the reason for the less significant differences observed here may lie in the way the data were compiled: it may have been a mistake to use qualitative data on the expression of TLR-2 as to whether it is present or not. In general, converting quantitative data into qualitative ones reduces sensitivity.
From Fig. 1 to Fig. 4, we are looking at the results of 12 samples, where the on/off ratio is plotted for each sample, resulting in n=8 or n=4. In fact, as far as Fig. S2A shows, the authors used flow cytometry. If we used this data and did a violin plot with measurements instead of on/off, the differences would be very clear. If you review the data in this way, you may come to a different conclusion. I would try this view at least once.
We added two graphics where a change to the violin plot is done; as you observe, the results are equal (left: Figure 1B; right: Figure 2C). We consider that due to the allele C group being only n=4, the more appropriate form to present results is graphics where the reader can observe the individual behavior per patient; it is helpful to identify individual profiles. Therefore, the new version maintains the previous graphic.

Reviewer 2 Report
Comments and Suggestions for Authors
This study investigated the relationship between the genetic variants rs3804099 and rs3804100 and the risk of severe COVID-19. They discovered that while the C allele does not increase the risk of severe COVID-19, it impairs the ability of TLR-2 to correctly respond to the SARS-CoV-2 spike protein.
1. For Figure 1, it would be helpful to explain the rationale for using LPS as a stimulant in the experimental context.
2. Figure 2, Did the authors explore whether spike proteins from different SARS-CoV-2 variants exhibit differential binding to TLR2 based on the two genetic variants?
Author Response
This study investigated the relationship between the genetic variants rs3804099 and rs3804100 and the risk of severe COVID-19. They discovered that while the C allele does not increase the risk of severe COVID-19, it impairs the ability of TLR-2 to correctly respond to the SARS-CoV-2 spike protein.
- For Figure 1, it would be helpful to explain the rationale for using LPS as a stimulant in the experimental context.
Thank you for your assertive comment; we clarified the aim of LPS as a stimulus. The new version indicated this point in methods (lines 291-295). We add reference 22.
- Figure 2, Did the authors explore whether spike proteins from different SARS-CoV-2 variants exhibit differential binding to TLR2 based on the two genetic variants?
This question is very relevant for clarifying whether genetic variants affect the affinity for the spike protein. However, we decided not to pursue this question for two reasons:
- a) The presence of the specific allele was not associated with the severity risk, leading us to believe that using different spikes does not significantly impact disease outcomes.
- b) Regarding TLR-2 expression, it was observed in the group with the T allele (the reference group), not the C allele, and specifically in one B-cell subset. While it could be interesting to design an experiment to evaluate the function of that subset and clarify its efficiency in producing antibodies via TLR2/spike recognition, this question is outside the scope of our main aim.
In response to this valuable comment, the new version includes a phrase referencing [19] the literature that shows other SARS-CoV-2 variants have a differential affinity for receptors (mainly TLR-4), though this is still unknown for TLR-2 variants (Lines 252–257).